# The Role of Minerals in the Optimal Functioning of the Immune System

**DOI:** 10.3390/nu14030644

**Published:** 2022-02-02

**Authors:** Christopher Weyh, Karsten Krüger, Peter Peeling, Lindy Castell

**Affiliations:** 1Department of Exercise Physiology and Sports Therapy, Institute of Sports Science, University of Giessen, 35394 Giessen, Germany; Christopher.Weyh@sport.uni-giessen.de; 2School of Human Sciences (Sport and Exercise Science), University of Western Australia, Crawley, WA 6009, Australia; peter.peeling@uwa.edu.au; 3Western Australian Institute of Sport, Mt Claremont, WA 6010, Australia; 4Green Templeton College, University of Oxford, Oxford OX2 6HG, UK; lindy.castell@gtc.ox.ac.uk

**Keywords:** magnesium, zinc, iron, selenium, deficiency, immune function, diet, supplementation

## Abstract

Minerals fulfil a wide variety of functions in the optimal functioning of the immune system. This review reports on the minerals that are essential for the immune system’s function and inflammation regulation. We also discuss nutritional aspects of optimized mineral supply. The supply of minerals is important for the optimal function of the innate immune system as well as for components of adaptive immune defense; this involves defense mechanisms against pathogens in addition to the long-term balance of pro- and anti-inflammatory regulation. Generally, a balanced diet is sufficient to supply the required balance of minerals to help support the immune system. Although a mineral deficiency is rare, there are nevertheless at-risk groups who should pay attention to ensure they are receiving a sufficient supply of minerals such as magnesium, zinc, copper, iron, and selenium. A deficiency in any of these minerals could temporarily reduce immune competence, or even disrupt systemic inflammation regulation in the long term. Therefore, knowledge of the mechanisms and supply of these minerals is important. In exceptional cases, a deficiency should be compensated by supplementation; however, supplement over-consumption may be negative to the immune system, and should be avoided. Accordingly, any supplementation should be medically clarified and should only be administered in prescribed concentrations.

## 1. Introduction

Minerals are vital components of our food. They fulfil a wide variety of functions, such as building materials for our bones, influencing muscle and nerve function, and regulating the body’s water balance [1]. They are also components of hormones and enzymes and other biologically active compounds. Some minerals also have an important role to play in the optimal functioning of the immune system. This concerns both the innate defense system and the adaptive immune response. Accordingly, the supply of minerals can influence susceptibility to infections, but it also has effects on the development of chronic diseases [2,3].

For most individuals, a balanced diet is sufficient to supply the body with sufficient amounts of the vital minerals. However, there are now a growing number of individuals who are at risk of mineral deficiency. These include people with chronic illnesses, older people, people who live on vegetarian or vegan diets, or women during pregnancy. Athletes can also be at risk of mineral deficiency if they eat a one-sided or reduced diet, for example, to achieve a specific weight. In this case, the appropriate composition and dosage of the individual minerals should be discussed with a qualified nutritionist, and attention paid to possible interactions with medication [4].

If people have sufficient knowledge about minerals and their importance, in addition to the corresponding foods that contain such minerals, a diet can easily be adjusted when a specific mineral might be lacking. The purpose of this article was to present the immunological function of some individual minerals, as well as to give advice on adequate nutrition with these minerals. We focused on minerals that have the most associations with immune function, based on the available literature.

## 2. Magnesium

### 2.1. General Physiological Function of Magnesium (Mg)

Mg is an essential biological element that is found in cells in a bound form. It is also the most abundant divalent cation in living cells, and it has numerous important tasks in regulatory cellular functions. Mg fulfils its task primarily through its binding to organic substances, such as proteins, nucleic acids, and nucleotides [5]. It has an important role in the activation of enzymes, membrane function, and intracellular signaling. The ion also represents an important cofactor for many enzymes. It is involved in the synthesis and replication of RNA and DNA, as well as the secretion of enzymes and hormones [6,7]. Mg plays an important role in a variety of metabolic processes, including oxidative phosphorylation and muscle contraction. Furthermore, magnesium stabilizes membrane structures and the membrane potential. It also modulates transmembrane movement of ions through the modulation of iron transporters [8]. Mg is found in high concentrations in the cells and matrix of bone. Furthermore, even larger amounts are found in muscle cells, soft tissues, blood serum, and erythrocytes. There are only slight concentration gradients in magnesium between the intracellular and extracellular space. Nevertheless, free magnesium ions influence the potential at the cell membrane. Intracellularly, magnesium is controlled primarily by an active transport system [6,9].

### 2.2. Immunological Role of Magnesium

Mg has numerous roles in the regulation of immunological functions, particularly in terms of the function of numerous immune cell subpopulations [10]. The diverse effects can be particularly illustrated by the nature of the structures of the innate immune system, the adaptive immune system, and the regulation of acute and chronic inflammatory processes [11].

#### 2.2.1. Magnesium and the Innate Immune System

Mg has effects on the acute phase response and the function of macrophages, for example, in their response to cytokines. In monocytes, it was shown that Mg supplementation reduced cytokine production after toll-like receptor (TLR) stimulation. This immunoregulatory function was due to an increased IĸBα level, which resulted in reduced nuclear factor kappa-light-chain-enhancer of activated B cell (NF-κB) translocation [12]. Rats with Mg deficiency have also shown peripheral neutrophilia, associated with increased phagocytosis and oxidative burst. In addition, there was a disturbance in the function of mast cells with regard to the secretion of histamine [13].

#### 2.2.2. Magnesium and Adaptive Immunity

Mg has a major influence on the development, differentiation, and proliferation of lymphocytes [14]. In this context, a lack of Mg is thought to interfere with the process of apoptosis. It is known that Fas-induced apoptosis is an Mg-dependent process. Mg-deficient mice have also shown earlier involution of the thymus, which had a negative effect on the T-cell pool [15]. The Mg^2+^ transporter TRPM7 seems to play a special role in the development of T cells. In cells lacking this transporter, and thus the Mg supply, developmental inhibition and early cell death occurred [16]. Accordingly, T-cell function seems to be dependent on an adequate supply of Mg, both at the level of the individual cell and with regard to the T-cell pool of an individual. Furthermore, several studies in lymphocytes have shown that Mg^2+^ from intra-/extracellular sources regulated phosphoinositide metabolism [17]. Hence, Mg might indirectly affect various immune cell functions, such as proliferation, which depend on proper substrate metabolism [18].

Recent studies provided evidence that bone resorption, which is increased during pro-inflammatory conditions, supports the metabolic function of lymphocytes. In this regard, it is generally assumed that bone, as a storehouse for phosphates and minerals, also represents a reserve for the bioenergetic activity of immune cells. Mg has an important role as a cofactor for various enzymes that play a role in phosphorylation, for example, in the phosphorylation cascades of glyoclysis and nucleotide polymerisation [18]. Accordingly, under conditions of acute inflammation, the released Mg is suggested to be favorable for the metabolic activity of T cells, which determine central functions such as proliferation.

#### 2.2.3. Immunoregulating Effects of Mg

Mg deficiency seems to favor an overfunction of the innate immune defense, with simultaneous deficiency of the adaptive immune defense. This might be one reason why some human studies have provided evidence that Mg deficiency is associated with the condition of chronic, low-grade inflammation [19]. This has been shown, for example, by an inverse relationship between (low) serum Mg concentrations and increased systemic C-reactive protein (CRP) levels [20]. The causal relationship is not entirely certain, although there is evidence from in vitro studies showing that Mg deficiency leads to increased production of interleukin (IL)-β and tumor necrosis factor (TNF)-α. Mg deficiency also resulted in increased aggregation of platelets, which had an impact on microvascular function. Corresponding to these findings, a recent metanalysis proved that Mg supplementation reduces serum levels of CRP [21]. However, these studies were carried out with small numbers of participants, suggesting that more research is needed.

In experimental animals, an Mg-reduced diet immediately caused a marked Mg deficiency, which provided numerous hallmarks of an altered proinflammatory status. Systemic IL-6 levels increased, acute phase proteins were released, and there was an increase in correlates of oxidative stress, such as thiobarbituric acid-reactive substances (TBARS). In parallel, the activity of superoxide dismutase (SOD) and catalase decreased. In addition, less glutathione was synthesized, a process which is Mg-dependent. One cause of the increased oxidative stress was presumably the release of the proinflammatory neuropeptide Substance P, which has been shown to occur with Mg deficiency [22]. Even a short-term Mg deficiency caused an increased synthesis of ceramides, which are known to induce NFkB and, accordingly, the release of various proinflammatory cytokines, such as TNF-α, IL-1β, and IL-6. Experimental hypomagnesemia also disrupted Ca^2+^ homeostasis, which enhanced the pro-inflammatory effect of Mg deficiency [23]. Furthermore, there is evidence that the microbiota is also affected by systemic Mg deficiency. In this deficiency state, the concentration of bifidobacteria in the intestine was reduced while the barrier function of the intestinal wall was disturbed, which led to increased expression of TNF-α and IL-6 in the intestine and liver [24].

### 2.3. Magnesium and Infectious Diseases

The importance of Mg in the context of infections lies, above all, in the close interaction between vitamin D metabolism and the importance of Mg as a cofactor. This implies that, if Mg is in short supply, less vitamin D can be formed from its precursors [25,26]. However, most of the data on immunodepression, and the connection to Mg status, come from animal experiments. These have confirmed that an Mg deficiency leads to multiple disturbances of the inflammatory response, which can affect the risk of infection [27].

### 2.4. Nutritional Aspects of Magnesium

The recommended daily amount of magnesium to be consumed varies according to age and gender. On average, it is about 300–400 mg for men and 270–310 mg for women, always depending on which guidelines are used as a basis. With age, this recommendation usually increases slightly. Some scientists recommend the consumption of up to 500 mg per day [28]. Natural sources of magnesium are mainly fruits and vegetables, as well as nuts, seeds, and whole grain products. Good food sources of Mg can be found in Table 1. Mg is a highly soluble mineral, and cooking processes can lead to a loss. It is assumed that only about 30-40% of ingested Mg is absorbed intestinally. An increased Mg requirement seems to be present in athletes during exercise, particularly after chronic training sessions. However, the extent to which supplementation is necessary here is controversial [3].

## 3. Zinc

### 3.1. General Physiological Function of Zinc (Zn)

The essential trace element Zn is crucial for many physiological processes in humans, and it is one of the most frequently studied factors in nutrition and health. It plays elementary roles as a regulator or coenzyme of more than 300 enzymes [29]. It is also a component of transcription factors and is involved in the synthesis of DNA and RNA, as well as proteins. Zn acts as an antioxidant and influences the stability of biological membranes and the arrangement of multiprotein complexes, such as the T-cell receptor. Furthermore, Zn regulates the formation of hormones, as well as their receptors [30,31,32]. The total Zn concentration in an adult is approximately 2–3 g, of which 85% is distributed within muscles and bones. During transport via the blood stream, the largest proportion is found in the red blood cells, stored by superoxide dismutase and carbonic anhydrase [33]. However, in plasma, Zn is 60% bound to albumin, with a total plasma concentration of 12–16 μM [34,35]. The balance of cellular Zn status is closely associated with proliferation as well as differentiation and apoptosis [36]. A significant role is attributed to the trace element in the maintenance of immune homeostasis [31]. Thus, it affects the functional capacity of cells in the innate and adaptive immune system. In addition, Zn has a regulatory effect on the production of cytokines, the activity of the complement system, and antibody production [37,38,39]. Therefore, a deficiency of this element can contribute to a disturbance in immune function and thus have a significant impact on health.

### 3.2. Immunological Role of Zinc

#### 3.2.1. Zinc and the Innate Immune System

Adequate Zn uptake is important for the functioning of both the innate and the adaptive immune system [40]. In the context of the innate immune response, Zn plays a central role in the activity of nicotinamide adenine dinucleotide phosphate (NADPH) oxidase of neutrophil granulocytes [41,42]. Therefore, a reduced formation of reactive oxygen species (ROS) with reduced killing ability could be the result of Zn deficiency [40]. In vivo studies have also shown that Zn deficiency induces reduced adhesion and chemotaxis of monocytes and neutrophil granulocytes, as well as impaired macrophage maturation and activity [43]. Zn also has a significant role in natural killer (NK)cells. Zn deficiency can induce a reduced NK cell count in the peripheral blood and lead to impaired functionality. In this context, reduced chemotaxis and lysis of virus-infected cells or tumor cells have been demonstrated [44,45].

#### 3.2.2. Zinc and Adaptive Immunity

With regard to the functions of the adaptive immune system, Zn has an important influence on the formation, maturation, and function of T cells [31,46]. The reason for this is that Zn is an important structural element of the hormone thymulin, which is produced by the epithelial cells of the thymus and mediates the maturation of pre-T lymphocytes into T lymphocytes [47,48]. Accordingly, Zn deficiency inhibits T-cell maturation in the thymus, which leads to marked thymic atrophy and reduced numbers of pre-T lymphocytes in animal models [49]. In humans, it has been shown that Zn deficiency can result in a reduced ability in T-cell proliferation or cytokine production (e.g., IL-2, IFN-γ) [50,51,52]. Another significant role of Zn is that it is also important in T-cell differentiation processes [53]. Studies that induced Zn deficiency showed a reduction in the number of CD4+ T cells, which resulted in a disproportion of the CD4+/CD8+ ratio [54]. A strongly decreased CD4+/CD8+ ratio, for example, below 1.5, is clinically considered to be an indicator of, or a cause of, immune dysfunction and thus prognostic for various diseases. Within the CD4+ cells, there can also be a disproportion of Th-1 to Th-2 cells. In this case, the Th-1 cells are in the foreground, which show a more significant decrease. Consequences are, for example, reduced Th1-mediated cytokine production of TNF-α, IL-2, or IFN-γ [54,55].

In addition to the significant influence on T-lymphocytes, Zn deficiency can result in reduced maturation of B cells. Importantly this can result in reduced antibody production [40].

#### 3.2.3. Immune Regulatory Effects of Zn

In addition to its effect on selective immune functions, Zn status is associated with the overall regulation of the immune system. Studies show that, in addition to adapted immune activation, increased oxidative stress and systemic inflammatory responses are induced by Zn deficiency [56,57,58]. Zn can influence both the production and the signaling of numerous inflammatory cytokines. Chronic inflammation is associated with disease processes, and these can be additionally negatively affected by a Zn deficiency. It has been shown that patients with systemic inflammatory diseases, such as rheumatoid arthritis, and a simultaneous Zn deficiency, can present an increased expression of IL-1β, IL-1α, and IL-6 compared to patients with a higher Zn intake [59,60]. In addition, long-term Zn deficiency appears to promote changes in the chromatin structures of the IL-1β and TNF-α promoters, which enable the expression of both genes [36]. Zn can thus be characterized as a trace element that is responsible for the inhibition of the production of pro-inflammatory cytokines and can positively influence disease processes. The mechanisms mentioned above could be the reason for this since cytokines are mainly produced by T-lymphocytes and macrophages. The reduction in ROS also plays an important role [61]. In addition, the literature indicates the role of Zn as a negative regulator of the NF-κB signaling pathway. The pathway regulates the genes that control apoptosis, innate and adaptive immune responses, and inflammatory processes: as a result, the expression of proinflammatory cytokines, such as TNF-α, IL-1β, or IL-6, are regulated [62]. One of the most important inhibitory mechanisms is based on how Zn affects the expression of the protein A20. A20 is a Zn finger protein known to be an anti-inflammatory protein that also negatively regulates the tumor necrosis factor receptor (TNFR)- and TLR-initiated NF-κB signaling pathways [56].

Furthermore, in vivo experiments have shown that regulatory T cells (Treg) were induced and stabilized by the addition of Zn [63,64]. The influence of Zn is thus far-reaching and yet specific to each cell type.

### 3.3. Zn and Infections

Studies in recent years have repeatedly shown that Zn has a significant influence on viral infections and it can positively influence, or even prevent, the course of disease. The mechanisms are manifold and concern the entry of virus particles, fusion, replication, translation of virus proteins, and further release in a number of viruses [65,66]. With regard to clinical efficacy, the results of a meta-analysis suggested that Zn supplementation at a dose of >75 mg/day significantly reduced the duration of colds [67]. A particularly vulnerable group in this context is the elderly. In the context of the ageing of the immune system, there is an increased susceptibility to infections and their severity. It was shown that after 12 months of Zn supplementation (45 mg elemental Zn-gluconate/day), the incidence of infections was significantly lower in a group of 55–87-year-old individuals. This was accompanied by an increase in the Zn concentration in the plasma and a reduced formation of TNF-α and oxidative stress markers [68]. In vitro results also provide evidence that Zn cations, in particular, have been shown to inhibit SARS coronavirus RNA polymerase (RNA-dependent RNA polymerase) by reducing viral replication [69]. These important findings show that Zn can be considered as an active agent in the treatment of COVID-19 [70].

These studies highlight the importance of Zn for immune function, but there are also conflicting results showing that high doses of Zn of 100–300 mg per day cause immune dysfunction, and associated health problems. In line with this, Deuster refers to a tolerable upper limit of Zn as being 40 mg/day. Overall, this reflects the complexity of this type of research [71].

### 3.4. Zinc and Nutrition

Despite a heterogeneous distribution in the body, Zn has no clear storage compartment, which is why an organism is dependent on a daily intake. According to estimates, 17 % of the world population is at risk of insufficient Zn intake [72]. In the USA, an estimated 15% of the population has an insufficient Zn intake. Among older people, this estimate rises to 35–45% [73,74,75]. In addition to inadequate dietary Zn intake, older people also have reduced Zn absorption [76]. The effects of Zn deficiency share many similarities with age-related immune dysfunction, including thymic atrophy, lymphopenia, impaired adaptive immunity, and increased susceptibility to infection [46,77]. Reference values for Zn intake vary by age and sex and, for adults, also by phytate intake. These recommended values vary from country to country. The US Food and Nutrition Board recommends an intake of 11 mg/day and 8 mg/day for adult men and women, respectively [78]. The German Nutrition Society recommends 1.5 mg of Zn per day for infants aged between 0 and 4 months. The recommended intake for adolescents between 15 and 19 years old is 11 mg of Zn per day for females and 14 mg of Zn per day for males. The recommended intakes for females aged 19 years and older with low, medium, and high phytate intakes, respectively, are 7 mg, 8 mg, and 10 mg of Zn per day; for males the values are 11 mg, 14 mg, and 16 mg of Zn per day, respectively [79]. Good sources of Zn include beef, pork, cheese, milk, and eggs. Vegetable Zn sources are nuts, e.g., cashew and pecan nuts, wheat, or rye sprouts. Other good food sources of Zn can be found in Table 2. Despite positive findings on the influence on immune homeostasis, additional supplementation of Zn should always be tailored to individual needs, as excessive supplementation can have the opposite effect.

## 4. Copper

The copper (Cu) in the body is very sensitively balanced, as Cu in small amounts is necessary for many physiological processes on the one hand, but too much Cu can be very harmful on the other. For example, Cu serves as a cofactor in the respiratory chain and is used accordingly in the transfer of electrons to oxygen. Furthermore, Cu is an important cofactor for oxidative balance [80]. The amount of Cu in the human body is normally only about 80 to 150 milligrams. The trace element is found mainly in the liver, but also in bones and muscles. From the depots, it passes into the blood as needed, with excess Cu being released by the liver into the bile. The largest part is excreted by the intestine, a smaller part via the urine. The normal values for Cu in the blood are between 74 and 131 µg/dL in relation to age and sex. The urine value can also serve as a reference, where the values in the 24 h urine are below 60 µg in adults [81]. Among the micronutrients in the serum, Cu is also of particular importance and associations with systemic immune activity [82].

### 4.1. Copper and Immunity

A certain availability of Cu plays an important role in maintaining immune competence. Conversely, a Cu deficiency leads to reduced humoral and cellular immune function. It has been shown that Cu-deficient mice have a smaller thymus and an enlarged spleen. There is also a neutropenia and lower numbers of T cells. Mitogen-induced T-cell proliferation is impaired, as is the function of B cells and NK cells [83,84]. Therefore, the enrichment of plasma Cu levels could enhance both innate and adaptive immunity in humans. In this context, Cu is discussed as a trace element with antiviral activities that can serve preventively and therapeutically against COVID-19 [85]. Conversely, Cu is also an essential nutrient for microbial pathogens, and the body can inhibit the growth of pathogens by limiting their Cu availability. From the perspective of the immune system, the human body’s Cu intake should be well balanced, as small amounts are sufficient for optimal immune function, while too much can have harmful functions at the same time [86].

### 4.2. Copper and Nutrition

Nutrition also affects the Cu balance of the body. It was demonstrated in balance studies that daily intakes, which are below 0.8 mg/day, are followed by a net Cu loss, while net gains are observed above 2.4 mg/day [81]. Interestingly, the intake of zinc and Cu seems to have a reciprocal relationship. When large amounts of zinc are consumed, less Cu enters the body through the intestine. This effect is used in the therapy of Wilson’s disease, a disease characterized by a Cu overload [82]. In contrast, Cu deficiency only occurs very rarely, for example in cases of pronounced intestinal diseases. However, this usually leads to extensive deficiency symptoms, such as anemia and bone damage [86]. Cu is not a routine parameter, since a high Cu level in the blood is not significant on its own. Elevated levels can be found in a variety of diseases, such as diabetes mellitus, infections, or cancer: however, it is suggested to have no effect on the course of the disease or treatment [81].

## 5. Iron

Iron (Fe) is an essential dietary mineral used to support vital human functions, such as erythropoiesis, cellular energy metabolism, and immune system development and function. Despite its importance, Fe deficiency that results in anemia is the world’s most common nutrient disorder, reported to impact ~25% of the global population; although the prevalence is much greater in specific populations such as females (both pregnant and non-pregnant) and children (~30–47%) [87].

### Effect of Iron on the Modulation of Immune Function

The role of Fe in immune system modulation is interestingly complex, encompassing several mechanisms that create a juxtaposition for the role and benefits of healthy Fe stores. For instance, Fe deficiency has been shown to impair B-cell proliferation, T-lymphocyte function, and adaptive antibody responses, which appear to be well corrected by Fe supplementation [88,89,90]. Similarly, impaired immunocompetence in children of low socioeconomic statuses has been shown to improve through oral Fe supplementation, which, after correction, resulted in decreased morbidity and more illness-free days among this specific population [91]. In contrast, however, evidence suggests that the indiscriminate use of Fe supplements in developing countries may increase morbidity and mortality from health issues such as malaria, diarrhoeal illness, and tuberculosis (for review see [92]). It is thought that these issues arise from an increase in Fe availability, which is used by the invading pathogens to proliferate and survive. Notwithstanding, it is also common that individuals with Fe overload conditions (such as hemochromatosis) are at a greater risk of developing infections [93], a result of the excess Fe available for invading microorganisms to use in their favor.

Given the role that Fe can play in both host defense and pathogen proliferation, innate human immunity has evolved to reduce the amount of Fe available to invading pathogens within hours of infection onset. In such instances, a state of hypoferremia occurs, whereby concentrations of Fe in extracellular fluid and plasma dramatically decrease to limit Fe availability for microbial invaders [94]. This process is driven by the inflammatory cytokine cascade to host invasion, stimulating an increase in circulating levels of the hepatic derived peptide, hepcidin, which functions to modulate and degrade the activity of the ferroportin Fe export channels [95]. Degradation of the ferroportin Fe export channels is effective in restricting Fe availability for pathogen use, since this process results in the sequestration of Fe within macrophages, enterocytes, and hepatocytes, which thereby reduces any extracellular Fe levels that could be used for microorganism growth [92]. This mechanism of Fe withdrawal appears to be an effective evolutionary process developed to starve any invading pathogens’ access to much needed Fe stores.

In summary, it appears that maintaining a healthy Fe status is beneficial to ensure optimal human function (i.e., oxygen transport and energy metabolism), and immunocompetence that allows an appropriate immune response to be mounted to invading pathogens. However, it is also clear that the body’s innate and immediate response to infection is to restrict extracellular Fe availability, which limits the impact of the host feeding and propagating the pathogen. Interestingly, it has previously been suggested that individuals who are compromised by illness or infection might consider temporary behavioral changes, such as decreasing Fe supplementation, or reducing the consumption of high Fe containing foods, in order to assist the body’s innate Fe withdrawal response [96]. Good food sources of Fe can be found in Table 3. However, greater research is certainly needed in this area to better understand the complexities of Fe status and immune function, with a focus on transient nutrition strategies to minimize Fe availability to invading microorganisms.

## 6. Selenium

Selenium (Se) is an essential trace element which maintains homeostasis in humans and animals. A total of 50% of total body selenium is contained in skeletal muscle. It is also an antioxidant that is incorporated into the selenoprotein glutathione peroxidase (GPx) [97]. Importantly, Rayman [98] states that the discovery of disease-associated polymorphisms in selenoprotein genes has drawn attention to the relevance of selenoproteins to health. For example, selenoproteins are essential to the actions of the thyroid gland, and GPx specifically protects the thyroid gland via the removal of excess hydrogen peroxide [99].

Adequate selenium is essential for immune system function. Individual selenoproteins regulate inflammation and immunity, and the mechanisms by which Se influences these processes is discussed in a comprehensive article by Huang et al. [100]. They also reviewed the effects of Se levels on anti-viral immunity, autoimmunity, sepsis, allergic asthma, and chronic inflammatory disorders. It is important to bear in mind that supplementation is likely to benefit only those who have inadequate intakes.

Although relatively rare, selenium deficiency is important. However, the extent of this was only recognized in the 1980s after it was established that a positive response to Se therapy was the only way in which Se deficiency could be measured accurately. Prior to that it was suggested that the standard clinical measurements of plasma and urinary levels did not give adequate results [101]. More recently Ashton et al. [102] and Behne et al. [103], have undertaken a review of Se status and measurements. This has become an important issue in relation to COVID-19 (see later).

Some Se levels in relation to Se status are given in Lewis’s work [104]. For example, Se toxicity occurs at intakes of >900 µg/day, whereas deficiency occurs at <19 µg/day. Average UK intake is 75 µg/day versus 93 µg/day in females in USA. Interestingly, 66% of female athletes were reported to have Se intakes considerably lower than the recommended RDA in France, versus only 23% of male athletes.

Selenium deficiency at an intake of <19 µg/day can give rise to reduced immune function, cardiomyopathy, skeletal muscle myopathy, osteoarthropathy, some cancers, and viral disease [98]. Selenium supplementation in subjects with low plasma Se (<1.2 µmol/L) augments T-cell mediated immune responses to an oral vaccine, and has been seen to result in more rapid clearance of the poliovirus and a reduced number of viral mutations [105].

Recent studies have shown that selenium deficiency (induced via diet) can activate increased inflammatory cytokines via the inducible nitric oxide synthase (iNOS)/NF-kB pathway in pig brains [106]. This inflammation was found to be mediated by heat shock proteins (HSP).

Steinbrenner et al. [107] highlighted the importance of maintaining high levels of Se supplementation in patients suffering from HIV and influenza A viral infections, as well as in those coinfected with HIV and mycobacterium tuberculosis.

In recent years, and since the advent of the pandemic, there have been suggestions that selenium may have a role in combatting the effects of coronavirus disease. Of particular interest is its impact on the symptoms and outcome of SARS-CoV-2 when Se is deficient [108] which, according to the authors, is commonplace in a significant proportion of the global population. The majority of studies on this aspect of Se have considered its nutrient effects in the context of it being one component among others that have roles in immune function, such as vitamins and free fatty acids (FFA).

Interestingly, Zhang [109] et al. refer to a study that observed higher Se status in surviving vs. non-surviving COVID-19 patients. They also described a linear relation between the cure rate of the patients and regional Chinese status.

The Se content of soil varies considerably depending on geographical location [98]. For example, the distribution of soil Se in China revealed that the Se content was the highest in black soil and rice paddy soil, and lowest in sandy soil. Clay minerals are the main components of black soil, which has the highest Se content in Northeast China. Chinese cabbage is an important source of Se. Enriching soil with Se and Zn significantly increased their accumulation in Chinese cabbage, particularly Zn [110].

The daily dietary Se intake has become lower in the UK, which is thought to be due to the declining use of North American wheat in bread making. Generally, US wheat has ten times more Se than UK wheat, attributed to the fact that soils from the wheat-growing belt of America are more enriched in Se to a similar order of magnitude [111].

Foods that contain Se include the following: oats, wheat, brown rice, sunflower seeds, Brazil nuts, and mushrooms; various kinds of meat, tuna fish, salmon, and shellfish. However, Se is particularly low in dairy products, fruit, and vegetables [104]. Other good food sources of Se can be found in Table 4.

In summary, Se is essential for maintaining efficient function of the immune system. However, it is also possible for excessive amounts of Se to have adverse effects in those who already have adequate Se status.

## 7. Conclusion

The significance of the described minerals for the normal function of the immune system is well highlighted by the scientific literature. Although mineral deficiencies are rare, there are numerous at-risk groups who should pay attention to ensure they receive a sufficient supply. A balanced diet is basically a good basis for this. In exceptional cases, a deficiency should be compensated by supplementation; however, it should also be cautioned that some mineral supplements can be over-consumed, which can ultimately lead to negative effects on the immune system. Therefore, any form of corrective nutrient supplementation should always be medically clarified and should only be consumed in prescribed concentrations.

## Figures and Tables

**Table 1 nutrients-14-00644-t001:** Amounts of magnesium (mg) per designated serving of food.

**Vegetable Sources**	**Magnesium in mg**
150 g cooked spinach (originally frozen)	91.5
60 g oats	77.4
20 g sunflower seeds	67.2
20 g pumpkin seeds	57
2 slices wholewheat bread	55
250 g cooked potato	52.5
150 g banana	45
150 mL sparkling water	30
**Animal Sources**	
100 g cooked halibut (dry heat)	30
150 g plain yoghurt (3.5% fat)	18
30 g 1 slice of gouda cheese (30% fat)	9.9
30 g rolled fillet of ham	8.7

**Table 2 nutrients-14-00644-t002:** Amounts of zinc (mg) per designated serving of food.

**Vegetable Sources**	**Zinc in mg**
6 tbsp (180 g) wildrice, cooked	3.85
200 g wholewheat pasta	2.63
60 g oats	2.19
100 g hummus from chickpeas	1.84
200 g fresh spinach, cooked	1.62
100 g firm tofu	1.45
100 g cooked kidney beans (originally dried)	1.2
200 g champignons, cooked	1.02
**Animal Sources**	
1 medium oyster, cooked	12.7
150 g pork, cooked	3.51
2 slices (60 g) of mountain cheese (45% fat)	3.06
1 cup (200 g) skimmed milk (1.5% fat)	0.86
150 g codfish, cooked (dry heat)	0.72

**Table 3 nutrients-14-00644-t003:** Amounts of iron (mg) per designated serving of food.

**Vegetable Sources**	**Iron in mg**
200 g chanterelles, steamed	11.6
150 g spinach, steamed	4.6
60 g pistachios	4.4
60 g cashews	3.8
150 g chard, steamed	3.6
150 g chickpeas, canned	3.3
100 g tofu	2.8
250 g green peas, steamed	2.5
**Animal Sources**	
125 g pork liver, cooked	24.4
125 calf liver, cooked	11.3
125 beef liver, cooked	9.7
150 g deer, cooked	5.1
150 g beef, cooked	3.9

**Table 4 nutrients-14-00644-t004:** Amounts of selenium (µg) per designated serving of food.

**Vegetable Sources**	**Selenium in µg**
250 g mushrooms	17.5
150 g oats	14.6
25 g nut mix (with 15 g walnuts and 2 Brazil nuts)	9.0
70 g plain rice, cooked	7.0
70 g dried lentils	6.9
1 egg	6.0
100 g pepper	4.3
250 g potato	3.8
200 g apple	2.8
**Animal Sources**	
70 g mackerel	27.3

## Data Availability

Not applicable.

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
