# Peer review of "The Role of Minerals in the Optimal Functioning of the Immune System"

_nutrients, 2022, doi:10.3390/nu14030644_

Round 1

Reviewer 1 Report

The article entitled ‘The Role of Minerals in the Optimal Functioning of the Immune System” is an interesting paper that summarizes very clearly the role of magnesium, zinc, iron and selenium in the immune system. Nevertheless, I found some aspects that would need be improved.

In general, why did not the authors include copper? Why did they include the four minerals? The reason of such selection should be included in the Introduction section.

Other suggestions:

  1. The abstract is just a short introduction. To improve it, the aim of the article and the main conclusions should be included.
  2. Section 2.1. includes a lot of information about the physiological role played by magnesium. There are only 3 references and one of them is an editorial. Could the authors revise/add these references?
  3. Lines 67-8: ‘This importance becomes particularly clear when there is an Mg deficiency’. I think that this is common for all the minerals, this could be moved to the Introduction section.
  4. Could the sections 2.3, 2.4 and 2.5 be changed to 2.2.1, 2.2.2 and 2.2.3?
  5. Line 73: why the use of lymphokines? I think it is better the use of cytokines, and in this case, when the authors referred to those secreted by macrophages, it sounds better cytokines.
  6. Line 76: reference 10 is a study focused in rats instead mice.
  7. Line 106: C-reactive protein (C in capital letter).
  8. Line 116: ‘TBARS’ is generally used (instead ‘TBRAS’)
  9. Line 121: does it refer to NFkB translocation? Please specify.
  10. Line 122: TNF abbreviation was already used in line 108, where it must be explained. Similarly, IL- should be explained here.
  11. Tables 1 and 2: it is strange that it has two titles: one after ‘Table number’ and another in the first line of the table. I think that the second one could be used instead the former. Moreover it could be used ‘Vegetable sources’ similarly as ‘Animal sources’.
  12. Section 3: why the authors do not apply Zn for zinc as before in magnesium (Mg) and later in selenium (Se)?
  13. Section 3: it would be better to use subsections to distribute the role of Zn in innate and acquired immunity. Then NK cells (lines 194-197) must be moved to innate immunity.
  14. Line 172 includes ‘reactive oxygen species’ that are defined as ROS in line 214. It would be better indicating the meaning of ROS here.
  15. Line 201: what does the meaning of ‘adapted’ refer to?
  16. Similarly, NF-kB is defined in lines 215-216 but is already used before. Even, the explanation of this pathway should be moved before, the first time that it appears.
  17. Line 222 defines TLR when it is used before.
  18. Line 233: insert a space after ‘of’.
  19. Line 247: insert a space after ‘40’.
  20. Section 4: similarly to sections 2 , 3 and 5, it should be ‘Iron’ and some subsections would be grateful.
  21. Section 4: why the authors do not apply Fe as before in magnesium (Mg) and later in selenium (Se)?
  22. Section 5: similarly to sections 2 and 3, some subsections would be grateful.
  23. Paragraphs in lines 342-351: put attention and use the Greek letter µ instead of ‘u’.

Author Response

Response to reviewers

The authors thank the reviewers for their time in assessing our manuscript. We feel the suggestions made have improved our paper, and are thankful for the opportunity to submit a revised script. Please find below a detailed response to each reviewer comment.

Sincerely,

Authorship team

Reviewer 1

In general, why did not the authors include copper? Why did they include the four minerals? The reason of such selection should be included in the Introduction section.

Our response: Reviewer 1's objection is correct. Due to his/her query about copper, we have integrated the mineral and created a separate section for it (section 4). We also added the information that we focused on the minerals where we found the most associations with immune function based on the available literature (line 46-47).

  1. The abstract is just a short introduction. To improve it, the aim of the article and the main conclusions should be included.

Our response: We have adapted the abstract accordingly: however, there is a limit of 200 words, which is very short and leaves little room for an adequate conclusion. The last 3 sentences are changed to conclusion

  1. Section 2.1. includes a lot of information about the physiological role played by magnesium. There are only 3 references and one of them is an editorial. Could the authors revise/add these references?

Our response: Thank you very much for pointing this out. We have added the references in the revised version of the manuscript.

  1. Lines 67-8: ‘This importance becomes particularly clear when there is an Mg deficiency’. I think that this is common for all the minerals, this could be moved to the Introduction section.

Our Response: Reviewer 1 is correct. We have deleted the very general sentence at that point. It is already mentioned in the introduction.

  1. Could the sections 2.3, 2.4 and 2.5 be changed to 2.2.1, 2.2.2 and 2.2.3?

Our Response: We agree with this comment and have adapted it for Mg and Zn.

  1. Line 73: why the use of lymphokines? I think it is better the use of cytokines, and in this case, when the authors referred to those secreted by macrophages, it sounds better cytokines.

Our Response: Reviewer 1 is correct. Lymphokines has now been changed to cytokines.

  1. Line 76: reference 10 is a study focused in rats instead mice.

Our response: Thank you for the advice. It was corrected.

  1. Line 106: C-reactive protein (C in capital letter).

Our response: Amendment made

  1. Line 116: ‘TBARS’ is generally used (instead ‘TBRAS’).

Our response: Amendment made

  1. Line 121: does it refer to NFkB translocation? Please specify.

Our response: Amendment made

  1. Line 122: TNF abbreviation was already used in line 108, where it must be explained. Similarly, IL- should be explained here.

Our response: Amendment made

  1. Tables 1 and 2: it is strange that it has two titles: one after ‘Table number’ and another in the first line of the table. I think that the second one could be used instead the former. Moreover it could be used ‘Vegetable sources’ similarly as ‘Animal sources’.

Our Response: We agree with the reviewer’s comment here. Accordingly, we have adapted the table heading.

  1. Section 3: why the authors do not apply Zn for zinc as before in magnesium (Mg) and later in selenium (Se)?

Our response: Amendment made

  1. Section 3: it would be better to use subsections to distribute the role of Zn in innate and acquired immunity. Then NK cells (lines 194-197) must be moved to innate immunity.

Our response: Amendment made

  1. Line 172 includes ‘reactive oxygen species’ that are defined as ROS in line 214. It would be better indicating the meaning of ROS here.

Our response: Amendment made

  1. Line 201: what does the meaning of ‘adapted’ refer to?

Our Response: The term ‘adapted’ is not appropriate here. Therefore, we deleted it.

  1. Similarly, NF-kB is defined in lines 215-216 but is already used before. Even, the explanation of this pathway should be moved before, the first time that it appears.

Our response: Amendment made

  1. Line 233: insert a space after ‘of’.

Our response: Amendment made

  1. Line 247: insert a space after ‘40’.

Our response: Amendment made

  1. Section 4: similarly to sections 2, 3 and 5, it should be ‘Iron’ and some subsections would be grateful.

Our response: Sub heading now added as requested

  1. Section 4: why the authors do not apply Fe as before in magnesium (Mg) and later in selenium (Se)?

Our response: Fe now used in place of iron after first use, where defined.

  1. Section 5: Similarly, to sections 2 and 3, some subsections would be grateful.

Our response: Amendment made

  1. Paragraphs in lines 342-351: put attention and use the Greek letter µ instead of ‘u’ –

Our response: Amendment made

Reviewer 2 Report

Dear Authors,

I have read the manuscript and it is of clinical interest.

I think that some points must be clarify:

1) please add methods used to identify the papers

2) please add a figure for each compound able to show the mechanism of action 

Author Response

Response to reviewers

The authors thank the reviewers for their time in assessing our manuscript. We feel the suggestions made have improved our paper, and are thankful for the opportunity to submit a revised script. Please find below a detailed response to each reviewer comment.

Reviewer 2

  • please add methods used to identify the papers

Our response: We thank reviewer 2 for pointing out the method of literature search. However, we have not underpinned this with a systematic approach, as this paper is a narrative review, and does not require any such methodological section. We have only justified the choice of minerals that have been discussed here. We very much hope that reviewer 2 can accept the handling of his comment. 

  • please add a figure for each compound able to show the mechanism of action 

Our response: The minerals described here are involved in a wide range of immune processes. As such, it is not possible to create a figure here that adequately depicts the mechanistic processes of each mineral. Furthermore, five separate figures would overwhelm the paper and is not feasible. We have therefore added tables for the most important minerals, showing which minerals are contained in which foods and in what quantities. We found this more informative as a supplement for the reader than a graphic that might then only cover partial aspects or become confusing.

Round 2

Reviewer 1 Report

Dear editor,

The authors modified accordingly all my suggestions. I believe that the paper can be accepted.

Best regards 

Reviewer 2 Report

Dear Authors,

I have read the manuscript and I have not further comments

best regards

Luca